# Preparation and Drug Release Profile of Chitosan–Siloxane Hybrid Capsules Coated with Hydroxyapatite

**DOI:** 10.3390/pharmaceutics14051111

**Published:** 2022-05-23

**Authors:** Yuki Shirosaki, Yasuyo Tsukatani, Kohei Okamoto, Satoshi Hayakawa, Akiyoshi Osaka

**Affiliations:** 1Faculty of Engineering, Kyushu Institute of Technology, 1-1 Sensuicho, Tobata-ku, Kitakyushu 804-8550, Japan; 2Graduate School of Natural Science and Technology, Okayama University, Okayama 700-8530, Japan; en421745@s.okayama-u.ac.jp (Y.T.); en19713@s.okayama-u.ac.jp (K.O.); akiosaka@okayama-u.ac.jp (A.O.); 3Faculty of Interdisciplinary Science and Engineering in Health Systems, Okayama University, Okayama 700-8530, Japan; satoshi@okayama-u.ac.jp

**Keywords:** chitosan–siloxane hybrid, capsule, hydroxyapatite coating, drug release

## Abstract

Chitosan is a cationic polymer that forms polymerized membranes upon reaction with anionic polymers. Chitosan−carboxymethyl cellulose (CMC) capsules are drug delivery carrier candidates whose mechanical strength and permeability must be controlled to achieve sustained release. In this study, the capsules were prepared from chitosan−γ-glycidoxypropyltrimethoxysilane (GPTMS)−CMC. The mechanical stability of the capsules was improved by crosslinking the chitosan with GPTMS. The capsules were then coated with hydroxyapatite (HAp) by alternately soaking them in calcium chloride solution and disodium hydrogen phosphate solution to prevent rapid initial drug release. Cytochrome C (CC), as a model drug, was introduced into the capsules via two routes, impregnation and injection, and then the CC released from the capsules was examined. HAp was found to be deposited on the internal and external surfaces of the capsules. The amount of CC introduced, and the release rate were reduced by the HAp coating. The injection method was found to result in the greatest CC loading.

## 1. Introduction

Regenerative medicine involves the treatment of tissues and organs that have been lost due to illness or injury [1]. Tissue regeneration requires scaffolds that allow cells to adhere, proliferate, and differentiate. Porous materials are widely used as scaffolding materials because their porous structure promotes cell migration, proliferation, and differentiation [2,3]. However, when large porous scaffolds are used for bigger tissue defects, the time taken for the cells to reach the center of the scaffold increases. Recently, bottom-up tissue engineering approaches using assembling have been established [4,5]. If the supported cells can accumulate three-dimensionally in a tissue defect, it is considered possible to repair the tissue at an early stage.

Capsules have been reported as one of methods used for assembling tissues prototypes. Peanparkdee et al. reported that capsules with a core–shell-type heterogenous structure could work as a reservoir for the drug [6]. To be suitable for supporting the survival and proliferation of cells, the capsule membranes must be permeable to allow the exchange of gases such as oxygen and carbon dioxide, the transport of nutrients and waste products, and the introduction and sustained release of bioactive molecules [4,7]. Furthermore, the membranes must be stable for a prolonged period without showing toxicity towards the supported cells or decomposing in vivo.

Generally, impregnation is used to load drug molecules into capsules [8]. However, as the amount of drug loaded by impregnation depends on the adsorption and permeation of the capsule wall, it is saturated below the maximum solubility of the drug in water. Microinjection has been used to deliver biomolecules such as proteins, peptides, and drugs directly into single cells [9,10]. This method can control the delivery dosage and the efficiency of transduction.

Chitosan is a cationic polymer that forms polymerized membranes upon reaction with anionic polymers such as CMC [11,12,13,14]. Kong et al. reported that chitosan−CMC hollow capsules could be prepared by fast and slow mixing to remove the dyes with different charges [12]. These capsules have membranes with a porous structure and showed semi-permeability. Roy et al. prepared chitosan−CMC capsules crosslinked with glutaraldehyde to improve their thermal stability [13]. Chitosan−CMC capsules with 100−300 nm pores were able to infiltrate capsules with dye molecules of relevant sizes, and equilibrium was established between the dye molecules on either side of the capsule membrane [14]. Chitosan−CMC-based capsules are candidates for future use as sustainable materials for drug or cell vehicles. The microinjection of drugs into chitosan−CMC capsules is also expected to increase the drug loading capacity. However, the strength of chitosan−CMC capsules membranes should be improved to enable them to withstand needle piercing.

In this study, the chitosan−CMC capsules were crosslinked using GPTMS to im-prove their mechanical strength as well as their permeability to drugs. Moreover, the capsules were coated with HAp using an alternate soaking process [15] with the aim of suppressing the initial burst release and reinforcing the membrane strength. GPTMS has epoxy groups and methoxysilane groups that crosslink the chitosan chains to control their swelling behavior, biodegradability, mechanical properties, and cytocompatibility [16]. The silanol groups derived from GPTMS provide the apatite nucleation sites for calcium ions [17]. The carboxy groups of CMC are also negatively charged and induce cation adsorption. HAp deposition is expected to occur on the surfaces of capsules. In addition, silanol groups from GPTMS can be condensed to form siloxane networks between the capsules, which is useful for assembling 3D structures. Cytochrome C was selected as a model drug for several growth factors [18] and its loading and release behavior was investigated using impregnation and microinjection.

## 2. Materials and Methods

### 2.1. Preparation of Chitosan–Siloxane Capsules

Chitosan powder (molecular weight = 50,000–190,000 Da, deacetylation = 75–85%; Sigma-Aldrich, St. Louis, MO, USA) was dissolved in 0.1 M aqueous acetic acid to obtain a 1% *w*/*v* chitosan solution. An appropriate amount of 0.1 M calcium acetate monohydrate ((CH_3_COOH)_2_Ca·H_2_O, Wako Pure Chemicas Corp., Osaka, Japan) aqueous solution was added to the chitosan solution. Then, γ-glycidoxypropyltrimethoxysilane (GPTMS, 97%, Alfa Aesar, Ward Hill, MA, USA) was added to the chitosan solution to give a range of molar ratios (chitosan:GPTMS = 1:0, 1:1, 1:2; the name of the sample is indicated as Ch, ChG1, and ChG2, respectively.). Additionally, the solutions were stirred at room temperature for 1 h. CMC (Nacalai Tesque, Inc., Kyoto, Japan) was dissolved in distilled water to obtain 0.5 % *w*/*v*. Finally, 1 μL of the CMC solution was dropped into the chitosan−GPTMS solution to form the capsules with approximately 1 mm in diameter.

### 2.2. HAp Coating of the Capsules

An alternate soaking process [15] was used to coat the surface of the capsules with a HAp layer. Aqueous CaCl_2_ solution (0.2 M) and aqueous Na_2_HPO_4_ solution (0.12 M) were prepared. The chitosan–GPTMS–CMC capsules (ChG2) were soaked in 0.2 M CaCl_2_ solution at 37 °C for 30 min. After washing with distilled water, the capsules were soaked in 0.12 M Na_2_HPO_4_ solution at 37 °C for 30 min and washed with distilled water. This alternate soaking method was repeated 3 times.

### 2.3. Characterization of the Capsules

The obtained capsules were observed using inverted microscopy (Eclipse. TS100, Nikon Corp., Tokyo, Japan). Their dimeters were measured from images of 30 capsules using Image J (v1.53, National Institutes of Health, Bethesda, MD, USA). The capsules were lyophilized using a freeze-dryer (FDU-506, EYELA, Tokyo, Japan) for characterization. The degree of crosslinking was evaluated using the ninhydrin assay to detect free amino groups present in the samples. Ground capsules were suspended in a solution of ninhydrin in buffer (L-8500 Set, Wako Chemicals Corp., Osaka, Japan) and shaken at 80 °C for 20 min in a shaking water bath (BW201/BF200, Yamato Scientific Co., Ltd., Tokyo, Japan). The optical density of the obtained supernatant was recorded at 570 nm using an ultraviolet-visible spectrometer (UV-2550, Shimadzu Corp., Kyoto, Japan). The degree of crosslinking was calculated using Equation (1):Degree of crosslinking (%) = (1 − A*_ChG_*/A*_Ch_*) × 100(1)
where A*_ChG_* and A*_Ch_* are the absorbance of chitosan–GPTMS capsules and chitosan capsules, respectively:

To examine their mechanical properties, the capsules were soaked in phosphate-buffered saline solution (PBS, pH 7.4) and shaken at 37 °C and 200 rpm for 14 days in a shaking water bath. The structural integrity of the capsules was then assessed using inverted microscopy. The damage rate was calculated using Equation (2).
Damage rate (%) = The number of broken capsules/The initial number of capsules × 100(2)

The freeze-dried capsules were pelleted by pressure molding. The crystal phases of the capsule pellets were examined using thin-film X-ray diffraction (TF-XRD, X’Pert-ProMPD, PANAlytical, Almelo, the Netherlands: CuKα, λ = 1.5406 Å, the fixed angle = 1.000°, 45 kV–40 mA, 2θ/θ scans, 0.02°/step with a count time of 4.00 s). The outer and inner surface morphologies of the capsules were observed using a scanning electron microscope (SEM, VE-9800, KEYENCE Corporation, Osaka, Japan) equipped with energy-dispersive X-ray spectroscopy (EDX, EDAX genesis, AMETEK Inc., Berwyn, PA, USA). The freeze-dried capsules were coated with Au/Pd to a thickness of approximately 5 nm by ion sputtering (E-1010, Hitachi High-Tech Corporation, Tokyo, Japan).

### 2.4. Drug Loading of the Capsules

Cytochrome C (Horse Heart, MW = 12,384, Nacalai Tesque, Inc., Kyoto, Japan) was selected as a model growth factor protein. Two methods, impregnation and injection, were used to load cytochrome C into the capsules. For impregnation systems, the capsules were incubated in 200 μg/mL of cytochrome C aqueous solution and kept at 37 °C for up to 28 h. To determine the maximum loading amount, the capsules were incubated in several concentrations of aqueous cytochrome C solution at 37 °C for 24 h. The absorbance of the supernatant was measured using a spectra scanning multimode reader (Varioskan Flash 5250040, Thermo Fisher Scientific, Waltham, MA, USA) at 528 nm and the loading was calculated. The injection system was set up as shown in Figure 1a. The ChG2 coated with HAp capsules were fixed on the hole (φ = 700 μm) of the glass chamber. Aqueous cytochrome C solution at different concentrations (0.3 μL) was directly injected into the capsules using a glass micro pipette (MP-010, φ = 10 μm, Micro Support Co., Ltd., Shizuoka, Japan) as shown in Figure 1b. The amounts of cytochrome C introduced were 9.3 μg (maximum amount by impregnation) and 27.9 μg (maximum solubility of cytochrome C in water) per capsule.

### 2.5. Drug Release

After loading the cytochrome C, the capsules were incubated in acidic saline solution (adjusted pH 3.0 by HCl) at 37 °C. The amount of cytochrome C released from the capsules into the saline solution was measured using a spectrum scanning multimode reader at 528 nm. The release profile was fitted using the Weibull function [19,20,21] as shown in Equation (3). This model has been used in drug release and dissolution studies [20]. *k* corresponds to the maximum amount of released drug. *a* and *b* refer to the specific release mechanisms with the diffusion coefficient of the matrices [21].
F(*t*) = *k*(1 − exp(−*at^b^*))(3)

### 2.6. Statistical Analysis

Testing of the mechanical stability and drug release test was carried out in triplicates. The results of the Weibull fitting were presented as means ± standard deviations (SD) and analyzed using one-way analysis of variance (ANOVA) followed by Tukey’s test or *t*-test with a significance level of *p* < 0.01.

## 3. Results and Discussion

The microscopy images are shown in Figure 2. The capsules were uniform in size with narrow distribution and spherical shape regardless of their composition. The capsule diameter was dependent on the volume of the CMC droplets in the chitosan−GPTMS solution as shown in Figure 3. Their theoretical diameter was calculated using Equation (4).
(4)$$D\;=2{\left(\frac{3V}{4\pi }\right)}^{\frac{1}{3}}$$

The measured capsule diameters were lower than the theoretical values regardless of the amount of GPTMS included. This difference is attributed to a strong electrostatic interaction between the amino groups of chitosan and the carboxy groups of CMC [12] as well as the crosslinking and siloxane networks formed by GPTMS. However, the results showed that the capsules of the desired size could be easily made by dropping CMC into chitosan−GPTMS. The larger capsules could be used as vehicles for the storage of a large amount of drug [12], while the smaller one could be used for cell encapsulation [4]. In this study, capsules with a diameter of approximately 1 mm were used in the injection method due to the convenience of the microinjection equipment. The degrees of crosslinking for ChG1 and ChG2 were 9.1 ± 2.7% and 27.3 ± 3.2%, respectively. The epoxy groups of GPTMS reacted with the amino groups in the chitosan chains, as previously reported [16,17]. Figure 4 shows the damage rate of the capsules after shaking in PBS. ChG2 showed a lower damage rate than either Ch and ChG1. This suggests that the crosslinking between chitosan and GPTMS and the siloxane network derived from GPTMS, led to the capsule membrane having a dense structure, which inhibited breakage. However, the damage rates of the capsules within 2 weeks were less than 10%, even for Ch and ChG1. The capsule membranes were therefore expected to be robust, even under body fluid conditions. Based on the damage rate results obtained, we chose to use ChG2 for HAp coating to test the drug release properties. Figure 5 shows the TF-XRD patterns of ChG2 capsules before and after coating with HAp using the alternate soaking process. The HAp-coated capsules showed new peaks assigned to HAp (ICDD JCPDS#09-0432) at 2θ = 26° and 33° in the pattern. Figure 6 shows SEM images of the ChG2 capsule before and after coating with HAp. The outer and inner surfaces of the capsules before coating were smooth and their thickness was around 1 μm. After alternate soaking, deposits of HAp were observed on both the outer and inner surfaces. This indicates that HAp did not completely cover the surface of the capsules. HAp deposits were only observed on the walls of the capsules only. During the alternate soaking process, the positively charged calcium ions (Ca^2+^) interacted with the negatively charged CMC [12] or silanol groups derived from GPTMS [16,17]. The phosphate ions (PO_4_^3^^–^) then reacted with the calcium ions to form nuclei for the growth of HAp [17]. Figure 7 shows a schematic illustration of the capsule structure.

Figure 8 shows the amount of cytochrome C loaded into the capsules. The loading of the ChG2 and ChG2 + HAp capsules reached equilibrium at 3 h and 16 h, respectively, as shown in Figure 8a. The cytochrome C molecule is a few nanometers in size and was immediately load into the ChG2 capsules. The pore size of the capsule wall was therefore thought to be more than a few tens of nanometers. The HAp deposits suppressed the loading rate, but the loading amount eventually reached the same as that for ChG2. The pore structure is thought to be retained in the areas without HAp deposits. From the results shown in Figure 8a, the immersion time was fixed at 24 h. Figure 8b shows the amount of cytochrome C loaded into the capsules as a function of the equilibrium concentration. The maximum loading of cytochrome C into ChG2 and ChG2 + HAp was 30 μg per capsule at 2000 μg/mL and 9.3 μg per capsule at 1000 μg/mL, respectively. More cytochrome C can be loaded into ChG2 than into ChG2 + HAp because of the relatively sparse wall network formed by only chitosan and GPTMS. Chitosan and HAp are stable at physiological condition, and the differences in their release are small in in vitro condition at pH 7.0. In this study, we used the conditions at pH 3.0 for an accelerated test. Moreover, we expect that the capsules can be used at the site of inflammation (low pH) to regenerate the tissues in the future. The release percentage of cytochrome C from ChG2 and ChG2 + HAp and the constants derived from the Weibull model are shown in Figure 9 and Table 1. The *b* value derived from the fitting of Equation (3) to the drug release curve data can be used to determine the release mechanism [19,20,21]. The *b* value for ChG2 was less than that of ChG2 + HAp. In the range 0.39 < *b* < 0.69, the release mechanism is diffusion in fractal or disordered substrates different from the percolation cluster. In contrast, the mechanism is diffusion in normal Euclidian substrates with the contribution of another release mechanism in the range 0.75 < *b* < 1. This indicates that the early release rate was reduced by the HAp deposits. In this study, the release was carried out at pH 3. Matsumoto et al. reported that the dissolution rate of HAp at pH 4 increased in a time-dependent manner, while the release of cytochrome C depended on the HAp dissolution [22]. After incubating for 1 h, the rate of release of cytochrome C from ChG2 + HAp was similar to that from ChG2 owing to the larger pores contributing to the dissolution of HAp.

The capsule wall of the ChG2 capsule was too weak for it to be injected by a glass needle; thus, we used only the ChG2 + HAp capsule for the injection method. Figure 10 shows the release percentage of cytochrome C from ChG2 + HAp after loading using the injection method. The Weibull model constants derived from Figure 10 are shown in Table 2 and Table 3. Seventy percent of the loaded cytochrome C was released within 6 h. Both b values were in the range 0.75 < b < 1. The loading method did not affect the release profile. The b value for 27.9 μg of cytochrome C was more than 1. The estimated release mechanism therefore resulted in a sigmoid curve, indicative of complex re-lease. The impregnation method was dependent on the permeability of the capsule wall or the adsorption on the wall surface. Therefore, the amount of cytochrome C loaded was lower than that for the injection method, which directly introduced cargo into the capsule. After loading cytochrome C by injection, the hole made by the glass micropipette remained, which enhanced the release of cytochrome C. The improvement of the capsule flexibility, or the complete recovery of the capsules is required. However, it is expected to be possible to uniformly load cells with drugs using the injection method.

## 4. Conclusions

Chitosan−GPTMS−CMC capsules were prepared based on an electrostatic interaction between the carboxy groups of CMC, and the amino groups of chitosan, and GPTMS crosslinking and siloxane network formation. The addition of GPTMS improved the mechanical properties of the capsule. The release of cytochrome C from the capsules was suppressed by coating the capsule wall with HAp. The amount of cytochrome C loaded into the capsules could be increased by using an injection method.

## Figures and Tables

**Figure 1 pharmaceutics-14-01111-f001:**
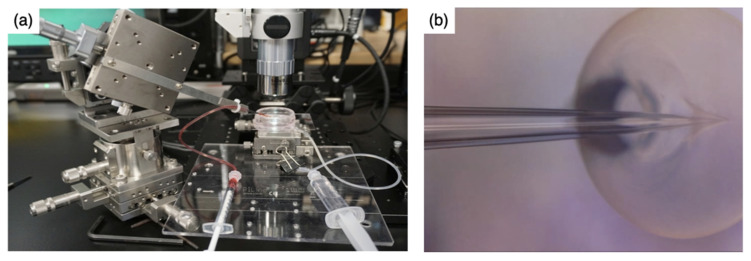
Photographs of the injection system. (**a**) The injection equipment; (**b**) injection into the ChG2 capsule using a glass micropipette.

**Figure 2 pharmaceutics-14-01111-f002:**
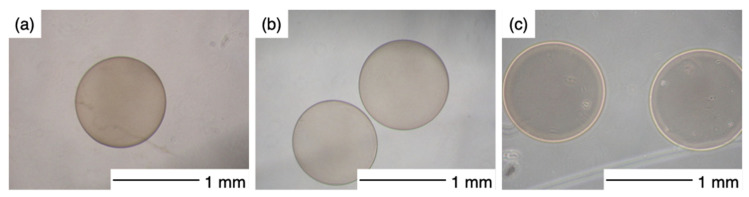
Microscopy images of chitosan–GPTMS–CMC capsules. (**a**) Ch, (**b**) ChG1, and (**c**) ChG2.

**Figure 3 pharmaceutics-14-01111-f003:**
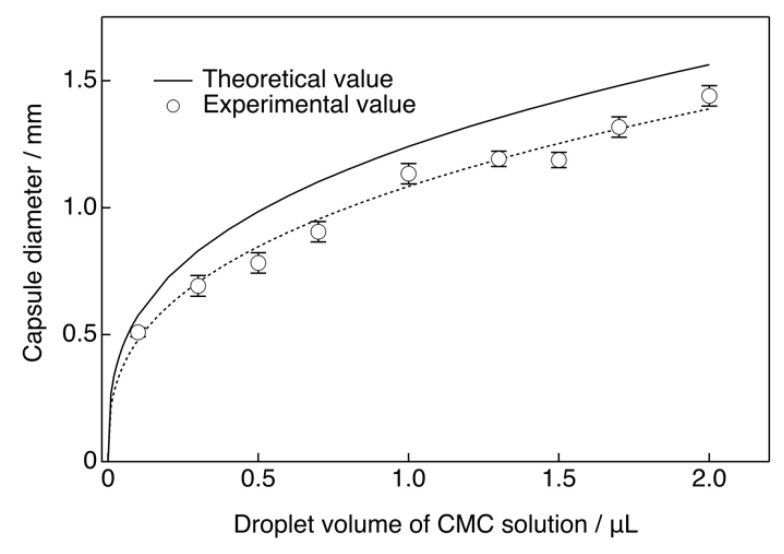
Theoretical and experimental diameters of the ChG1 capsules.

**Figure 4 pharmaceutics-14-01111-f004:**
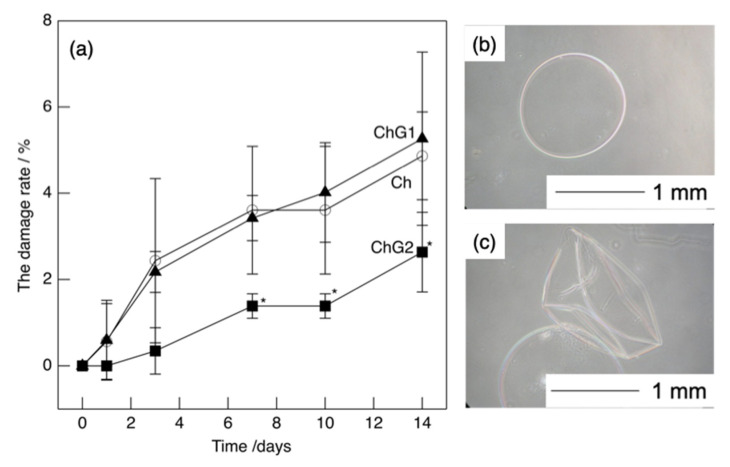
The damage rate of the chitosan–GPTMS–CMC capsules (**a**) and microscopy images of the ChG2 capsules before shaking (**b**), and after damage at 14d (**c**). * *p* < 0.01.

**Figure 5 pharmaceutics-14-01111-f005:**
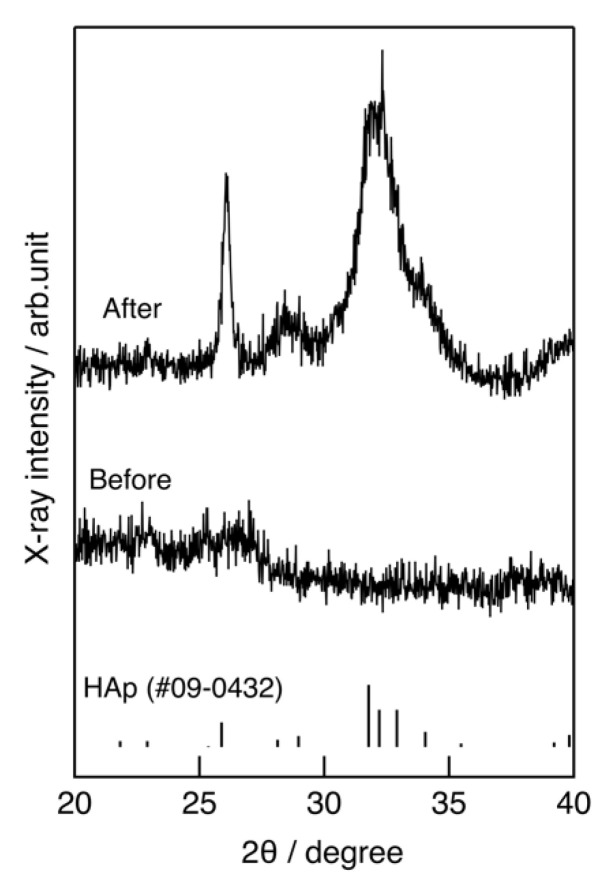
TF-XRD patterns of the ChG2 capsules before and after coating with hydroxyapatite.

**Figure 6 pharmaceutics-14-01111-f006:**
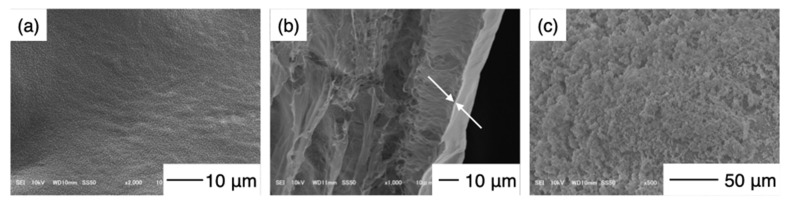
SEM images of the ChG2 capsule surface. (**a**) The outer surface and (**b**) the cross-section. The arrows indicate the wall of the capsule. (**c**) The outer surface following HAp deposition.

**Figure 7 pharmaceutics-14-01111-f007:**
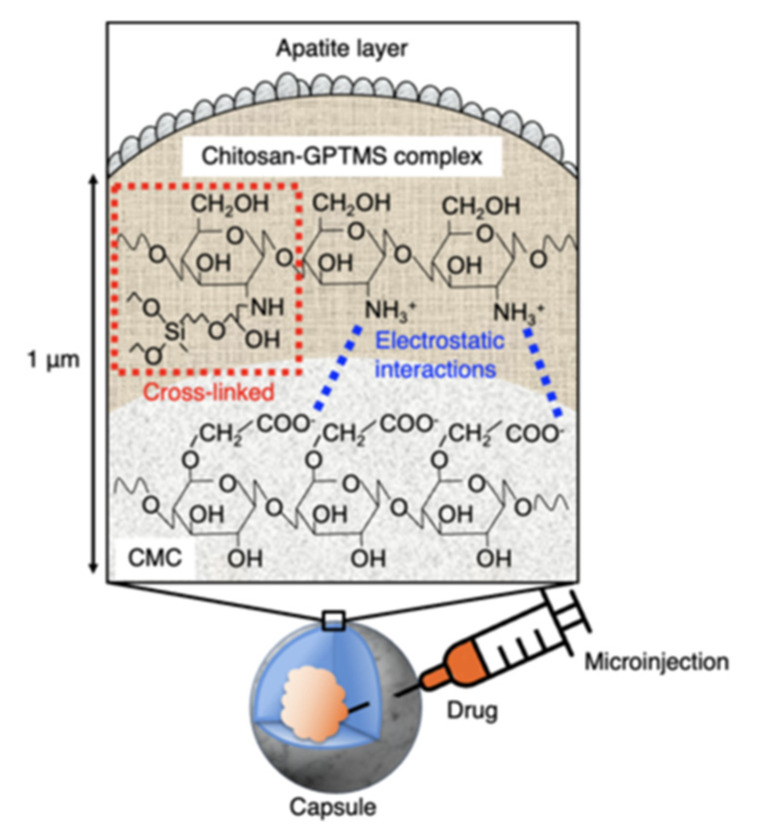
A schematic illustration of the capsule structure.

**Figure 8 pharmaceutics-14-01111-f008:**
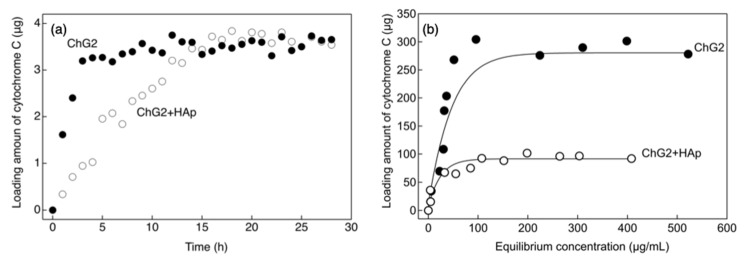
Amount of cytochrome C loaded into the capsules by the impregnation method. (**a**) Per capsule as a function of time at an equilibrium concentration of 200 μg/mL, (**b**) per ten capsules as a function of equilibrium concentration at 24 h.

**Figure 9 pharmaceutics-14-01111-f009:**
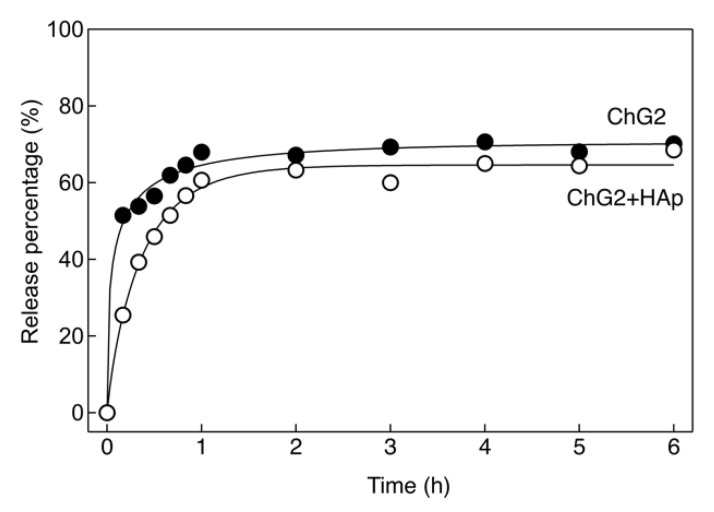
Release percentage of cytochrome C induced by the impregnation method from ChG2 and ChG2 + HAp as a function of time.

**Figure 10 pharmaceutics-14-01111-f010:**
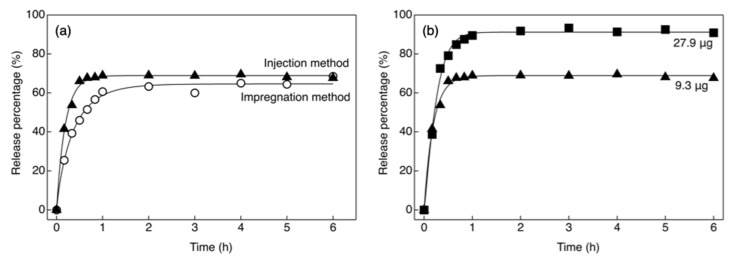
Release percentage of cytochrome C from ChG2 + HAp as a function of time: (**a**) impregnation and injection method; (**b**) different amount of cytochrome C loaded using injection method.

**Table 1 pharmaceutics-14-01111-t001:** Weibull equation constants derived from Figure 9.

Sample	*a*	*b*	*k*
ChG2	2.55 ± 0.25	0.41 ± 0.05	70.42 ± 1.01
ChG2 + HAp	2.33 ± 0.05	0.84 ± 0.03 *	64.60 ± 0.09 *

* *p* < 0.01.

**Table 2 pharmaceutics-14-01111-t002:** Weibull equation constants derived from Figure 10a.

Sample	*a*	*b*	*k*
Impregnation	2.33 ± 0.05	0.84 ± 0.03	64.60 ± 0.09
Injection	5.07 ± 0.05 *	0.98 ± 0.01 *	69.04 ± 0.21 *

** p* < 0.01.

**Table 3 pharmaceutics-14-01111-t003:** Weibull equation constants derived from Figure 10b.

Loading Amount (μg)	*a*	*b*	*k*
9.3	5.07 ± 0.05	0.98 ± 0.01	69.04 ± 0.21
27.9	5.12 ± 0.20	1.20 ± 0.03 *	90.97 ± 0.23 *

* *p* < 0.01.

## Data Availability

The data that support the findings of this study are available from the corresponding authors, upon reasonable request.

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
