# Peer review of "Preparation and Drug Release Profile of Chitosan–Siloxane Hybrid Capsules Coated with Hydroxyapatite"

_pharmaceutics, 2022, doi:10.3390/pharmaceutics14051111_

Round 1

Reviewer 1 Report

The work "Preparation and drug release profile of chitosan−siloxane hybrid capsules coated with hydroxyapatite" has an actual and important subject of the research field.

The authors  were prepared  chitosan−γ-glycidoxypropyltrimethox- 15
ysilane (GPTMS)−carboxymethyl cellulose (CMC) capsules and coated them with hydroxyapatite and compared the release behavior of the drug loaded using the impregnation and injection method. Hydroxyapatite was found to be deposited on the internal and external surfaces of the capsules. The amount of CC introduced, and the release rate were reduced by the hydroxyapatite coating. The injection method resulted in the greatest CC loading.

The procedures and the methods were adequate to this study.

The results are interesting and useful.

The conclusions were based on the obtained experimental results.

The references covering the specific research field.

However, a little polish of the figures were welcome (ex. fig. 6 the resolution for 6a and b and the contrast for 6c).

Author Response

We wish to thank the reviewer for comments. In accordance with the reviewer’s
comment, we have changed Figure 6.

Reviewer 2 Report

The reviewed article “Preparation and drug release profile of chitosan−siloxane hybrid capsules coated with hydroxyapatite” by Yuki Shirosaki et al. contains a number of interesting and worth publishing results in the field of research on drug delivery pathways. Nevertheless, I suggest the Authors consider extending the article by several issues:

1) Fig. 3 compares the theoretically estimated and experimentally determined capsule diameter. The first doubt is raised by the experimental point in (0,0) - has such a measurement really been made? The explanation of the discrepancy between the theoretical and experimental value of the capsule diameter is also insufficiently justified (the differences reach even 20%?). Perhaps it is worth considering proposing a correction of a simple theoretical model so as to obtain a more reliable representation of the experimental data? Then, it would probably be possible to indicate correction factors taking into account the factors responsible for the shrinkage of capsules indicated by the authors and to determine their impact, also taking into account the size of the capsules.

2) Does (and if so, how much) capsule filling change the internal pressure?

3) What is the tension of the capsule wall? its stiffness? extensibility?

4) How uniform is the thickness of the capsule wall? and does the release process (mechanism, time) depend on the thickness of the capsule wall? is the thickness of the wall the same regardless of the capsule size?

5) The Authors of the article use the term "pore size" many times in relation to the porosity of the capsule walls. Is the pore size in question known? Examined? Describing it as "a few tens of nanometers" seems insufficient in view of the key role these pores play in drug release. What is the number of these pores? Are they uniformly sized structures? Are the pores of the same size regardless of the capsule size?

6) Do HAp close or only narrow (and to what diameter) pores?

7) Is it possible to determine the time constants of the processes shown in Figs. 8 and 9?

8) P.7, l. 214: The last sentence suggests that the mechanism of injecting (and implicitly releasing) a drug in liquid and solid form will be the same - is that what the Authors mean?

9) Please note that on the list of 20 cited papers, there are as many as 5 articles by the Authors.

Minor editorial notes:

1) Already in chapter 2.1. it is worth indicating what capsule size Authors are talking about.

2) P.3. l. 95, "... Health ..."

3) I suggest move the position of equation 1 and put it just after the words: "... using equation".

4) It is incomprehensible why the Authors use a verbal notation of size instead of symbols in the equations, eg. Eq 4: "The theoretical diameter" instead of "d".

Author Response

  • Fig. 3 compares the theoretically estimated and experimentally determined capsule diameter. The first doubt is raised by the experimental point in (0,0) - has such a measurement really been made? The explanation of the discrepancy between the theoretical and experimental value of the capsule diameter is also insufficiently justified (the differences reach even 20%?). Perhaps it is worth considering proposing a correction of a simple theoretical model so as to obtain a more reliable representation of the experimental data?
    Then, it would probably be possible to indicate correction factors taking into account the factors responsible for the shrinkage of capsules indicated by the authors and to determine their impact, also taking into account the size of the capsules.

We appreciate the reviewer’s comment on this point. The experimental point in (0,0)
was the error and we have changed Figure 3. Regarding, the differences of the
diameter between theoretical and experimental values, we explained by the
electrostatic interaction and crosslinking (page6, line170).

2. Does (and if so, how much) capsule filling change the internal pressure?

We thank the reviewer for this comment. We would not be unable to do this
additional study because of no equipment to measure the internal pressure.

3. What is the tension of the capsule wall? its stiffness? extensibility?

We thank the reviewer for this comment. After hydroxyapatite coating, the stiffness
seems to be increasing. And the measurement is important for future application with
microinjection. However, we would not be unable to do this additional study because
of no equipment to measure the capsule wall tension.

4. How uniform is the thickness of the capsule wall? and does the release process
(mechanism, time) depend on the thickness of the capsule wall? is the thickness of the wall
the same regardless of the capsule size?

We appreciate the reviewer’s comment on this point. We agree that the thickness is
related with the release process. However, the thickness of the capsule wall does not
depend on the capsule size in this study.

5. The Authors of the article use the term "pore size" many times in relation to the
porosity of the capsule walls. Is the pore size in question known? Examined? Describing it
as "a few tens of nanometers" seems insufficient in view of the key role these pores play in
drug release. What is the number of these pores? Are they uniformly sized structures? Are
the pores of the same size regardless of the capsule size?

We wish to thank the reviewer’s for this comment. We have observed the capsule
surface by SEM, but it was difficult to get clear image at high magnification by our
equipment. The existence of the pore and the size is expected from the release result.
(page7, line201).

6. Do HAp close or only narrow (and to what diameter) pores?

We wish to thank the reviewer’s for this comment. HAp coating was not completed
and got the pores to be narrow from the release results.

7. Is it possible to determine the time constants of the processes shown in Figs. 8 and 9?

We agree that additional information on Figure 8 and 9 as the reviewer suggested
would be valuable. When we use log-log plot, we can get some other constants.
However, because we could not explain more new things and consider that this may
not be necessary.

8. P.7, l. 214: The last sentence suggests that the mechanism of injecting (and implicitly
releasing) a drug in liquid and solid form will be the same - is that what the Authors mean?

In according with the reviewer’s comment, we have changed this to “uniformly load
drug with cells” (page8, line236).

9. Please note that on the list of 20 cited papers, there are as many as 5 articles by the
Authors.

We agree with the relevance of the references, and have uploaded.

Minor editorial notes:
1. Already in chapter 2.1. it is worth indicating what capsule size Authors are talking about.

Accordingly, we have added “with approximately 1 mm in diameter” (page2, line92).

2. P.3. l. 95, "... Health ..."

This error has been corrected in accordance with the reviewer’s comment. (page3,
line103)

3. I suggest move the position of equation 1 and put it just after the words: "... using
equation".

In accordance with the reviewer’s comment, we have changed this position. (page3,
line113)

4. It is incomprehensible why the Authors use a verbal notation of size instead of symbols
in the equations, eg. Eq 4: "The theoretical diameter" instead of "d".

In accordance with the reviewer’s comment, we have changed this point. (page5,
line168)

Reviewer 3 Report

The study claimed that the capsules were prepared from chitosan-γ-glycidoxypropyltrimethoxysilane (GPTMS)−CMC and coated with hydroxyapatite.  The release behavior of the drug loaded by the impregnation and injection method were compared using cytochrome C as a model drug.The results showed that the release of cytochrome C from the capsules was suppressed by coating the capsule wall with hydroxyapatite. The amount of cytochrome C loaded into the capsules could be increased by using an injection method. I think the paper should be accepted after minor revision. My questions and suggestions are as follows:

1. There are many reports about chitosan capsules, and the hydroxyapatite coating is very common, please give your innovative points in the introduction section.

2. The microinjection of drug into capsulecould lead to hole on the surface of capsule, how large the hole is? And how the hole would affect the release behavior of the drug loaded in capsule?

3. P75-82 : “Chitosan powder (MW = 50,000–190,000 Da, DA = 75–85%; Sigma-Aldrich, St. Louis,  MO, USA) was dissolved in 0.1 M aqueous acetic acid to obtain a 1 mass/v% chitosan” “Carboxymethyl cellulose (CMC, Nacalai Tesque, Inc., Kyoto, Japan)  was dissolved in distilled water to obtain 0.5 mass/v%. “  The concentration unit “mass/v%” is not a scientific expression. Do you mean “Chitosan solution 1%(mass/v), Carboxymethyl cellulose solution 0.5 % (mass/v)”? Please check it carefully.

Author Response

  1. There are many reports about chitosan capsules, and the hydroxyapatite coating is
    very common, please give your innovative points in the introduction section.

We thank the reviewer for this comment. In this study, we have focused on and
showed the possibility the microinjection method to load the drug in the capsules.
Microinjection systems were used for cells to transfer substances into cells. We could
not find the application of microinjection for drug loading into the capsules and have
suggested in this study. Hydroxyapatite coating was used to improve the mechanical
strength of the chitosan-CMC capsule membrane for the injection. We confirm the
references the reviewer suggested; however, they are for the chitosanhydroxyapatite composite bulk materials for bone defects. Thus, we have updated
some references relate with the capsules.

2. The microinjection of drug into capsule could lead to hole on the surface of capsule,
how large the hole is? And how the hole would affect the release behavior of the drug
loaded in capsule?

We appreciate the reviewer’s comment on this point. In this study, we used a glass
micro pipette with φ = 10 μm and a hole size after injection seems to be same. This
remained hole is only one but beggar than the pores from the polymer networks on
the capsule wall. For future work, we would like to improve the elastic of the capsule
wall to close the hole after injection.

3. P75-82 : “Chitosan powder (MW = 50,000–190,000 Da, DA = 75–85%; Sigma-Aldrich,
St. Louis, MO, USA) was dissolved in 0.1 M aqueous acetic acid to obtain a 1 mass/v%
chitosan” “Carboxymethyl cellulose (CMC, Nacalai Tesque, Inc., Kyoto, Japan) was
dissolved in distilled water to obtain 0.5 mass/v%. “The concentration unit “mass/v%” is not a
scientific expression. Do you mean “Chitosan solution 1%(mass/v), Carboxymethyl cellulose
solution 0.5 % (mass/v)”? Please check it carefully.

In accordance with the reviewer’s comment, we have changed mass/v% to “%w/v”.
(page2, line85, 92)

Reviewer 4 Report

In the present manuscript, chitosan-CMC-based capsules, crosslinked with glycidoxypropyltrimethoxysilane and coated with hydroxyapatite, were proposed and investigated as drug delivery carriers for Cytochrome C.

Some following issues should be carefully considered to further improve the quality of this paper:

  • I suggest to reorganize the introduction by moving the drug loading methods (lines 51-56) either before discussing about chitosan and merging with the 2nd paragraph, or after GPTMS part.
  • There are some inadvertencies: In figure 7a, the maximum loading capacity after 24 hours is practically the same for both types of capsules (ChG2 and ChG2+HAp), however in figure 7b, where the same capsules were exposed for 24 hours to cyt C solutions of different concentrations, there is a big difference in the drug loading amount when the same 200 ug/mL solution was used. Moreover, in text, the maximum loading capacity of the ChG2 and ChG2+HAp capsules were reported to be reached at 2000 ug/mL and 1000 ug/mL, respectively, but in the figure 7b the maximum concentration used was 500 ug/mL. Also, as shown in figure 7b, those maximum loading amounts were attained starting with the 100 ug/mL solution.
  • What was the rationale for choosing the pH=3 as the releasing medium from a pharmacological point of view?
  • The caption of figure 7 should be appended with the drug loading method used (impregnation method).
  • The abbreviations should be clearly stated at the first appearance in text, e.g.: Hap
  • The abbreviations of the synthesized capsules (Ch, ChG1, ChG2, ChG2+HAp) are not presented in a clear way. They should appear in the Materials and Methods section.
  • Phrases on lines 31-32, 56-57, 94 needs to be corrected.
  • English should be modified by a native speaker

In conclusion, this manuscript is suitable for publication, but not in this state because of the poor English and needs a major revision.

Author Response

Some following issues should be carefully considered to further improve the quality of this
paper:
1. I suggest to reorganize the introduction by moving the drug loading methods (lines
51-56) either before discussing about chitosan and merging with the 2nd paragraph, or after
GPTMS part.

In accordance with the reviewer’s comment, we have moved the loading method
before 2nd paragraphs.

2. There are some inadvertencies: In figure 7a, the maximum loading capacity after 24
hours is practically the same for both types of capsules (ChG2 and ChG2+HAp), however in
figure 7b, where the same capsules were exposed for 24 hours to cyt C solutions of different
concentrations, there is a big difference in the drug loading amount when the same 200
ug/mL solution was used. Moreover, in text, the maximum loading capacity of the ChG2 and
ChG2+HAp capsules were reported to be reached at 2000 ug/mL and 1000 ug/mL,
respectively, but in the figure 7b the maximum concentration used was 500 ug/mL. Also, as
shown in figure 7b, those maximum loading amounts were attained starting with the 100 ug/mL solution.

We wish to thank the reviewer for this comment. Fig 7(a) showed the result per a
capsule and Fig 7(b) showed one per ten capsules because the measurement at low
concentration. To make this point clearer, we have explained in the figure caption.

3. What was the rationale for choosing the pH=3 as the releasing medium from a
pharmacological point of view?

We wish to thank the reviewer for this comment. Chitosan and hydroxyapatite are
stable at physiological condition and the release differences are a little in vitro
condition at pH 7.0. In this study, we used the condition at pH 3.0 as an accelerated
test. Moreover, we expect that these capsules can be used at inflammation site (low
pH) to regenerate the tissues for future.

4. The caption of figure 7 should be appended with the drug loading method used
(impregnation method).

In accordance with the reviewer’s comment, we have added “impregnation method”
to the caption of Figure 7.

5. The abbreviations should be clearly stated at the first appearance in text, e.g.: Hap

Accordingly, we have added the abbreviation “(HAp)” (page1 line17) and changed some
points. (page1, line20, 22, page2 line63, 72, 93, page3, line94, page4, line143, page6,
line186, 187, 188, 190, page9, line246)

6. The abbreviations of the synthesized capsules (Ch, ChG1, ChG2, ChG2+HAp) are
not presented in a clear way. They should appear in the Materials and Methods section.

Accordingly, we have added a sentence, “The name of the sample is indicated as Ch,
ChG1, and ChG2, respectively.” (page2, line89)

7. Phrases on lines 31-32, 56-57, 94 needs to be corrected.

These errors have been corrected in accordance with the reviewer’s comment.

8. English should be modified by a native speaker

The paper has been edited and rewritten by an experienced scientific editor, who has
improved the grammar and stylistic expression of the paper. We have added information
to “Acknowledgement”.

Reviewer 5 Report

The aim of the manuscript was to prepare chitosan−GPTMS−CMC capsules and coat them with hydroxyapatite. The coated and uncoated capsules were characterized, and in the case of the coated capsule, the drug loading methods (impregnation and injection methods) were compared.

The methods used to characterize the capsules are well chosen and provide sufficient support for the conclusion, but the presentation and statistical evaluation of the results are incomplete in some cases, thus I have the following questions and suggestions:

  1. Abbreviations used for sample names are not indicated in the method section (Ch, ChG1, ChG2 ...).
  2. Why did you use a release medium with a pH of 3, is it suitable for any application?
  3. How many parallel measurements were used for each method.
  4. Why did you use the Weibull model? Please indicate R2 values in order to present the fitting of the Weibull model.
  5. No SD values are given in the drug release curves.
  6. A statistical evaluation should be used to compare drug release curves.
  7. Release rate or released amount of the drug was presented in the Figure 8 and 9? I think it is not a rate value.

Author Response

  1. Abbreviations used for sample names are not indicated in the method section (Ch,
    ChG1, ChG2 ...).

Accordingly, we have added a sentence, “The name of the sample is indicated as Ch,
ChG1, and ChG2, respectively.” (page2, line88)

2. Why did you use a release medium with a pH of 3, is it suitable for any application?

We wish to thank the reviewer for this comment. Chitosan and hydroxyapatite are
stable at physiological condition and the release differences are a little in vitro
condition at pH 7.0. In this study, we used the condition at pH 3.0 as an accelerated
test. Moreover, we expect that these capsules can be used at inflammation site (low
pH) to regenerate the tissues for future.

3. How many parallel measurements were used for each method.

We carried out the drug release test in triplicates. In accordance with the reviewer’s
comment, we have added the statistical analysis. (page5, line158)

4. Why did you use the Weibull model? Please indicate R2 values in order to present
the fitting of the Weibull model.

We acknowledgement the reviewer’s comment on this point. Weibull model has been
demonstrating in drug release and dissolution studies. The fitting is nonlinear
regression and use of R2 values in nonlinear is not standard and we could not show it.
We have added a sentence with a reference [20].

This model has been demonstrating in drug release and dissolution studies [20].
(page4, line154)
20. Corsaro, C.; Neri, G.; Mezzasalma, M.A.; Fazio, E. Weibull modeling of controlled
drug release from Ag-PMA Nanosystems. Polymers 2021, 13, 2897-2913.

5. No SD values are given in the drug release curves.
6. A statistical evaluation should be used to compare drug release curves.

We carried out the drug release test in triplicates. In accordance with the reviewer’s
comment, we have added the statistical analysis. (page5, line158)

7. Release rate or released amount of the drug was presented in the Figure 8 and 9? I
think it is not a rate value.

These errors in Figure 8 and 9 have been corrected in accordance with the
reviewer’s comment.

Reviewer 6 Report

The topic proposed by the authors is interesting. In this study, the authors have detailed the design, characterization and the drug release profile of chitosan−siloxane hybrid capsules coated with hydroxyapatite.

The introduction section is too weak and should be improved. Uniform the information given in the introduction and in the discussion of the results. The discussion must refer to the obtained results with possible correlation to the literature, while the introduction should contain all the topics relevant for the discussion.

The discussion section should be restructured and inserted as a distinct part of the manuscript. It is brief and conducted incorrectly. There was a lack of greater comparisons of the results obtained in this research with other similar studies, and preferably in the last 5 years. The authors should emphasize the advantages of using and compare them with others chitosan-hydroxyapatite hybrid systems mentioned in the literature. There are communicated recent results about the characterization, the in vitro and in vivo researches on the effects of various chitosan–hydroxyapatite hybrid composite and their medical applications as drug delivery systems.

(see:

  • Jarquin-Yáñez K et al. Cellulose-chitosan-nanohydroxyapatite hybrid composites by one-pot synthesis for biomedical applications. Polymers 2021, 13, 1655.
  • Marouf N et al. Physicochemical properties of chitosan–hydroxyapatite matrix incorporated with Ginkgo biloba-loaded PLGA microspheres for tissue engineering applications. Polymers and Polymer Composites 2020, Vol. 28(5) 320–330.
  • Réthoré G et al. Silanization of Chitosan and Hydrogel Preparation for Skeletal Tissue Engineering. Polymers 2020, 12, 2823);
  • Dreghici DB et al. Chitosan–hydroxyapatite composite layers generated in radio frequency magnetron sputtering discharge: from plasma to structural and morphological analysis of layers. Polymers 2020, 12, 3065.
  • Chuanlei Ji et al. Salvianolic Acid B-Loaded Chitosan/hydroxyapatite scaffolds promotes the repair of segmental bone defect by angiogenesis and osteogenesis. International Journal of Nanomedicine 2019:14 8271–8284.
  • Li B et al. Biological and antibacterial properties of the micro-nanostructured hydroxyapatite/chitosan coating on titanium. Sci Rep 2019; 9, 1–10.
  • Cord-Landwehr S et al. Quantitative mass-spectrometric sequencing of chitosan oligomers revealing cleavage sites of chitosan hydrolases. Chem. 2017, 89, 2893–2900).

Some other aspects were found in this manuscript:

- different fonts were used in the text and in the figures;

- figures 3, 5 and 8 have a low accuracy;

- the abbreviations should be mentioned in the text (line 75 – MW, DA; line 87 – CaCl2, Na2HPO4; 103 –Ch; line 108 – PBS, line 122 – Au/Pd; line 168 – Hap; figure 2 – ChG1)

- the explanation of the abbreviation should be used only once in the text and should not be repeated, in order to decongest the text and facilitate the understanding of the information transmitted (carboxymethyl cellulose – lines 1, 82, 43)

- the authors should mandatory upgrade the references;

- spelling check of the text is mandatory.

- English including grammar, style and syntax, should be improved through the professional help from English Editing Company for Scientific Writings.

- a schematic representation of the study would be appreciated.

Author Response

The introduction section is too weak and should be improved. Uniform the information given
in the introduction and in the discussion of the results. The discussion must refer to the
obtained results with possible correlation to the literature, while the introduction should
contain all the topics relevant for the discussion.

We wish to express our deep appreciation to the reviewer for his insightful comment
on this point. We reconstructed Introduction and discussion with appropriate
references. Highlights in yellow were the added or modified sentences or references.

The discussion section should be restructured and inserted as a distinct part of the
manuscript. It is brief and conducted incorrectly. There was a lack of greater comparisons of
the results obtained in this research with other similar studies, and preferably in the last 5
years. The authors should emphasize the advantages of using and compare them with
others chitosan-hydroxyapatite hybrid systems mentioned in the literature. There are
communicated recent results about the characterization, the in vitro and in vivo researches
on the effects of various chitosan–hydroxyapatite hybrid composite and their medical
applications as drug delivery systems.

We thank the reviewer for this comment. In this study, we have focused on and
showed the possibility the microinjection method to load the drug in the capsules.
Microinjection systems were used for cells to transfer substances into cells. We could
not find the application of microinjection for drug loading into the capsules and have
suggested in this study. Hydroxyapatite coating was used to improve the mechanical
strength of the chitosan-CMC capsule membrane for the injection. We confirm the
references the reviewer suggested; however, they are for the chitosan-hydroxyapatite
composite bulk materials for bone defects. Thus, we have updated some references
relate with the capsules.

Some other aspects were found in this manuscript:
1. different fonts were used in the text and in the figures;

The error has been corrected in accordance with the reviewer’s comment.

2. figures 3, 5 and 8 have a low accuracy;

In accordance with the reviewer’s comment, we had changed Figure 3, 5, and 8 to
high resolution images.

3. the abbreviations should be mentioned in the text (line 75 – MW, DA; line 87 – CaCl2,
Na2HPO4; 103 –Ch; line 108 – PBS, line 122 – Au/Pd; line 168 – Hap; figure 2 – ChG1)

Accordingly, we have changed MW, DA, and PBS to “Molecular weight,
Deacetylation, and Phosphate buffer saline solution (PBS, pH7.4)”, and the name of
the sample was indicated. (page2, line82, 88, page3, line117) CaCl2, Na2HPO4, and
Au/Pd are chemical formula and symbol, and we did not change.

4. the explanation of the abbreviation should be used only once in the text and should
not be repeated, in order to decongest the text and facilitate the understanding of the
information transmitted (carboxymethyl cellulose – lines 1, 82, 43)

Accordingly, we have changed carboxymethyl cellulose to only “CMC”. (page2,
line90)

5. the authors should mandatory upgrade the references;

In accordance with the reviewer’s comment, we have changed some references.

6. spelling check of the text is mandatory.

As requested, we have checked spelling.

7. English including grammar, style and syntax, should be improved through the
professional help from English Editing Company for Scientific Writings.

The paper has been edited and rewritten by an experienced scientific editor, who has
improved the grammar and stylistic expression of the paper. We have added
information to “Acknowledgement”.

8. a schematic representation of the study would be appreciated.

As requested, we have added the schematic presentation of the study as Figure 10.

Reviewer 7 Report

Concerning the manuscript “Preparation and drug release profile of chitosan−siloxane hybrid capsules coated with hydroxyapatite”

There is no doubt that this article contains valuable information about the release behaviour of hydroxyapatite-coated Chitosan-GPTMS-CMC capsules preloaded with cytochrome c by impregnation and injection methods. Its quality is enough to be published without any correction.

It has been a pleasure to read it.

Author Response

We appreciate the reviewer’s comments.

Reviewer 8 Report

This is a moderately interesting paper dealing with synthesis and characterization of chitosan-carboxymethyl cellulose capsules crosslinked with γ-glycidoxypropyltrimethoxysilane. Authors also utilized hydroxyapatite coating to reduce initial burst release of the encapsulated drug. A model drug cytochrome C was loaded into capsules via impregnation and injection. According to the authors, coating with hydroxyapatite reduced release rate of cytochrome C from the capsules. However, several major revisions have to be addressed carefully before any further decision regarding this manuscript can be made.

  1. The Introduction section is not clear. How the first paragraph is related with the next part of the section? I suggest the authors should clearly explain the aim of the article and the field where they plan to use the results described in the manuscript
  2. Please provide also a description of model drug cytochrome C. Why this drug was used as a model? How cytochrome C is related with regenerative medicine?
  3. Authors should include section Materials in the manuscript
  4. In line 76 please change 1 mass/v% to 1% w/v. The same recommendation is for the rest of the manuscript
  5. The lyophilization process should be transferred to section 2.2, as this is a description of a preparation rather than analysis. Was any cryoprotectant used?
  6. What buffer was used for ninhydrin assay?
  7. Why the release study was performed in acidic pH? What buffer was used? How the samples were incubated and withdrawn? All this questions should be discussed in the article
  8. There is no any statistical evaluation in the article. How much replicates were performed in Fig. 4 and 8?
  9. What does Ch and ChG1 stand for?
  10. A comparison of theoretical and experimental diameters provided only for Why authors selected only ChG1?
  11. The discussion of results of capsules’ diameters is poor. Why this experiment is important and what a conclusion should the reader draw?
  12. The synthesis of capsules is not visualized in the results. Please provide a synthesis scheme and sketch synthesis mechanism briefly.
  13. In Fig. 4C at what day microscopy evaluation was used?
  14. In Fig. 8 how cytochrome C was loaded into capsules? I suggest Fig 8 and 9a to be overlapped.
  15. Authors characterized loading as 30μg per capsule in the text and 7 μg per capsule in Fig. 9b. Please unify this characteristic
  16. There is no discussion on a and k constants of Weibull function. Please explain the meaning of each and provide a comparison of coated and uncoated capsules.
  17. Further directions are unclear. Please discuss your plans regarding improvement the capsules and their practical usage.
  18. The evaluation of the osteoblasts proliferation should be performed to prove the biocompatibility and efficacy of the capsules.

Author Response

  1. The Introduction section is not clear. How the first paragraph is related with the next
    part of the section? I suggest the authors should clearly explain the aim of the article and the
    field where they plan to use the results described in the manuscript

We wish to express our deep appreciation to the reviewer for his insightful comment
on this point. We reconstructed Introduction and discussion with appropriate
references.

2. Please provide also a description of model drug cytochrome C. Why this drug was
used as a model? How cytochrome C is related with regenerative medicine?

We appreciate the reviewer’s comment on this point. Cytochrome C was selected as
the model growth factor protein in drug delivery system. We have added the
information of cytochrome C with reference “Cytochrome C was selected as model
drug for several growth factors [18].” in Introduction (page2, line77).
18 Jongoaibookit, L.; Franklin-Ford, T.; Murphy, L.W. Mineral-coated polymer
microspheres for controlled protein binding and release, Adv. Mater. 2009, 21, 1960-
1963.

3. Authors should include section Materials in the manuscript

We appreciate the reviewer’s comment. However, we have given all material
information in Materials and Methods.

4. In line 76 please change 1 mass/v% to 1% w/v. The same recommendation is for the
rest of the manuscript

In accordance with the reviewer’s comment, we have changed mass/v% to “%w/v”.
(page2, line84, 91)

5. The lyophilization process should be transferred to section 2.2, as this is a
description of a preparation rather than analysis. Was any cryoprotectant used?

We appreciate the reviewer’s concerns on this point. However, we used the
lyophilization process to only the samples for structural analysis. And we did not used
the cryoprotectant.

6. What buffer was used for ninhydrin assay?

We used the ninhydrin assay kit (L-8500 Set, Wako Chemicals Corp, Japan). The
buffer was included lithium acetate dihydrate and propylene glycol monomethylether.
The details (concentration, etc.) are not open from the company and we have added
only kit information.

7. Why the release study was performed in acidic pH? What buffer was used? How the
samples were incubated and withdrawn? All this questions should be discussed in the article

We wish to thank the reviewer for this comment. Chitosan and hydroxyapatite are
stable at physiological condition and the release differences are a little in vitro
condition at pH 7.0. In this study, we used the condition at pH 3.0 as an accelerated
test. Moreover, we expect that the capsules can be used at inflammation site (low
pH) to regenerate the tissues for future. We have added the details of the
experiments in Methods section.

8. There is no any statistical evaluation in the article. How much replicates were
performed in Fig. 4 and 8?

We carried out the drug release test in triplicates. In accordance with the reviewer’s
comment, we have added the statistical analysis. (page5, line158, Figure 4, Table 1,
2, 3)

9. What does Ch and ChG1 stand for?

Accordingly, we have added a sentence, “The name of the sample is indicated as Ch,
ChG1, and ChG2, respectively.” (page2, line88)

10. A comparison of theoretical and experimental diameters provided only for Why
authors selected only ChG1?

We appreciate the reviewer’s comment on this point. We also examined the
diameters for Ch and ChG2. However, the capsule diameters do not depend on
composition, and we selected only one composition.

11. The discussion of results of capsules’ diameters is poor. Why this experiment is
important and what a conclusion should the reader draw?

We thank the reviewer for this comment. To make this point clearer, we have added
“However, the results showed that the capsules with the desired size can be easily
made by drop of CMC into chitosan−GPTMS. The bigger sized capsules can be
applicable to vehicle as storage with high amount of drug and release [12], and the
smaller one can be used cells capsulation [4].”.(page6, line172)

12. The synthesis of capsules is not visualized in the results. Please provide a synthesis
scheme and sketch synthesis mechanism briefly.

As requested, we have added the schematic presentation of the study as Figure 10.

13. In Fig. 4C at what day microscopy evaluation was used?

We thank the reviewer for this comment. We have added the day to Figure 4 caption.

14. In Fig. 8 how cytochrome C was loaded into capsules? I suggest Fig 8 and 9a to be
overlapped.

We agree that this point requires clarification. Fig 8 is that cytochrome C was induced
by impregnation method and compared with and without hydroxyapatite coating.
Figure 9 is the results of the capsules with hydroxyapatite coating because injection
method can be used for it. We have added the information in Figure 8 and in
Methods. (page4, line142)

15. Authors characterized loading as 30μg per capsule in the text and 7 μg per capsule
in Fig. 9b. Please unify this characteristic

For microinjection method we used only ChG2+HAp capsules because ChG2 was
too weak to inject by the glass needle. 30 μg per capsule is for ChG2 and the results
in Fig.9 is for ChG2+HAp. We agree that this point requires clarification and have
changed the following text.

“The amounts of cytochrome C introduced into the capsules was 9.3 μg (maximum
amount by impregnation) and 27.9 μg (maximum solubility of cytochrome C in water)
per capsule.” was changed to “The amounts of cytochrome C introduced was 9.3 μg
(maximum amount by impregnation) and 27.9 μg (maximum solubility of cytochrome
C in water) per capsule.” (page4, line148)

“From the results of Figure 7(a), the immersion time was fixed at 24h.” (page7,
line204)

“The capsule wall of ChG2 capsule was too weak to inject by the glass needle and
we used only ChG2+HAp capsule for injection method.” (page8, line223)

We have deleted the following text “The equilibrium loading amounts were
approximately 3.8 μg per capsule.”. Because we used Figure 7 to decide the
immersion time and did not use it for release test.

16. There is no discussion on a and k constants of Weibull function. Please explain the
meaning of each and provide a comparison of coated and uncoated capsules.

We acknowledgement the reviewer’s comment on this point. Weibull model has been
demonstrating in drug release and dissolution studies. Regarding R2 values, we
could not get them after fitting of Weibull model and the results showed the chi
square. We have added a sentence “This model has been demonstrating in drug
release and dissolution studies [20].” and “k corresponds to maximum released drug
amount. a and b are related to the specific release mechanisms with the diffusion
coefficient of matrices [21]” with a reference [20, 21]. (page4, line154)

20. Corsaro, C.; Neri, G.; Mezzasalma, M.A.; Fazio, E. Weibull modeling of controlled
drug release from Ag-PMA Nanosystems. Polymers 2021, 13, 2897-.2913.

21. Kosmidis, K.; Macheras, P. Monte Carlo simulations for the study of drug release
from matrices with high and low diffusivity areas. Int. J. Pharm. 2007, 343, 166–172.

17. Further directions are unclear. Please discuss your plans regarding improvement the
capsules and their practical usage.

We appreciate the reviewer’s concerns on this point. However, we have written in our
original manuscript about for future improvement. (page8, line236)

18. The evaluation of the osteoblasts proliferation should be performed to prove the
biocompatibility and efficacy of the capsules.

We agree that additional information on the biocompatibility for the osteoblasts as the
reviewer suggested would be valuable. Regrettably, however, because of
improvement of injection system under the sterile condition, we are unable to do the
experimentation now.

Others

We have changed “release rate” to “release percentage”. (page7, line210, page8,
line224, and Figure8 and 9)
We deleted the following parts.
these properties (page2, line63)
Shirosaki et al. investigated chitosan−GPTMS (γ-glycidoxypropyltrimethoxysilane) to
produce membranes and porous scaffolds that showed cytocompatibility with
osteoblastic cells and supported nerve regeneration [13−16]. (page2, line66)
the mechanical properties of the capsule wall were also improved. In this study, we
prepared chitosan−GPTMS−CMC capsules and coated them with hydroxyapatite and
compared the release behavior of the drug loaded using the impregnation and
injection method. (page2, line75)

Round 2

Reviewer 4 Report

The paper may be published in the present form.

Author Response

We wish to thank the reviewer for this comment.

Reviewer 6 Report

The authors mostly responded to the comments and suggestions and the manuscript was revised accordingly.

I consider it could be accepted for publication in Pharmaceutics journal.

Author Response

(The authors gave the same response as above.)

Reviewer 8 Report

Thank you for considering my comments. I noticed several improvements after revision, however, some points lack in the article.

  1. Please include Figure 10 in the manuscript
  2. Provide full explanation of in vitro release condition, that you responded to my comment, in the text of the article

Author Response

Thank you for considering my comments. I noticed several improvements after revision, however, some points lack in the article.

  1. Please include Figure 10 in the manuscript

Accordingly, we have added as Figure 7 and a sentence “Figure 7 shows a schematic illustration of the capsule structure.” (page7, line197), and changed figures No. after Figure 7.

  1. Provide full explanation of in vitro release condition, that you responded to my comment, in the text of the article

In accordance with the reviewer’s comment, we have added the explanation “Chitosan and hydroxyapatite are stable at physiological condition and the release differences are a little in vitro condition at pH 7.0. In this study, we used the condition at pH 3.0 as an accelerated test. Moreover, we expect that the capsules can be used at inflammation site (low pH) to regenerate the tissues for future.”(page8, line212).